# On the Developmental Timing of Stress: Delineating Sex-Specific Effects of Stress across Development on Adult Behavior

**DOI:** 10.3390/brainsci8070121

**Published:** 2018-06-29

**Authors:** Anna Schroeder, Michael Notaras, Xin Du, Rachel A. Hill

**Affiliations:** 1Department of Psychiatry, School of Clinical Sciences, Monash University, Clayton 3168, Australia; anna.schroeder@monash.edu (A.S.); xin.du@monash.edu (X.D.); 2Center for Neurogenetics, Brain & Mind Research Institute, New York, NY 10065, USA; mjn2004@med.cornell.edu

**Keywords:** stress, animal models, development, sex, behaviour, HPA-axis

## Abstract

Stress, and the chronic overactivation of major stress hormones, is associated with several neuropsychiatric disorders. However, clinical literature on the exact role of stress either as a causative, triggering, or modulatory factor to mental illness remains unclear. We suggest that the impact of stress on the brain and behavior is heavily dependent on the developmental timing at which the stress has occurred, and as such, this may contribute to the overall variability reported on the association of stress and mental illness. Here, animal models provide a way to comprehensively assess the temporal impact of stress on behavior in a controlled manner. This review particularly focuses on the long-term impact of stress on behavior in various rodent stress models at three major developmental time points: early life, adolescence, and adulthood. We characterize the various stressor paradigms into physical, social, and pharmacological, and discuss commonalities and differences observed across these various stress-inducing methods. In addition, we discuss here how sex can influence the impact of stress at various developmental time points. We conclude here that early postnatal life and adolescence represent particular periods of vulnerability, but that stress exposure during early life can sometimes lead to resilience, particularly to fear-potentiated memories. In the adult brain, while shorter periods of stress tended to enhance spatial memory, longer periods caused impairments. Overall, males tended to be more vulnerable to the long-term effects of early life and adolescent stress, albeit very few studies incorporate both sexes, and further well-powered sex comparisons are needed.

## 1. Introduction

Within the neurosciences, stress is a ubiquitous phenomenon studied across multiple models, at various levels of analysis and with differing methods. One commonality in the study of stress is the understanding that it is an experiential phenomenon that is cognitively framed by perceptual cues, responded to via coping mechanisms, and is controlled by the reactivity of neuroendocrine systems.

Acute stress allows the body to recruit the energy and resources to cope with the situation. The persistence of stress (chronic stress), however, can have a hazardous effect on the body, increasing the risk of schizophrenia, depression, heart disease, and a variety of other illnesses [1,2]. The acute stress response, or “fight and flight” response, was first described by Walter Bradford Cannon early in the 20th century. Perceived stressors which challenged “homeostasis”, a term Cannon coined, activate the sympatho-adrenomedullary system, resulting in the immediate release of catecholamines. As the name suggests, the immediate effect of the acute response is to marshal the body’s resources to combat, or flee from, the immediate source of stress, by elevating heart rate, releasing stored energy to muscles, dilating blood vessels to muscles, and boosting metabolic rate [3]. In the brain, noradrenaline increases arousal and attention, as well as mediates memory formation and retrieval [4]. While noradrenaline has been shown to be necessary for attentional set-shifting [5], elevated levels have been shown to impair lexical-semantic and associative network flexibility [6]. Adaptation of the sympatho-adrenomedullary system occurs if the same stressor is repeatedly and predictably experienced, but upon novel stressors, the system reverts to high release of catecholamines [7,8]. Importantly, evidence shows that this acute stress response is also influenced by the slower central stress response system, the hypothalamic–pituitary–adrenal (HPA) axis [9]. Given the prevalent role of the HPA axis in regulating stress adaptation, its dysfunction has been heavily implicated in a wide variety of disorders [10].

The HPA axis modulates a multitude of bodily functions to adapt an organism to stressors in the environment. It self-regulates via a negative feedback loop. The HPA axis is activated by hypothalamic parvocellular neurons within the paraventricular nucleus, secreting corticotrophin-releasing factor (CRF) into the medial eminence, which in turn, triggers the release of adrenocorticotropic hormone (ACTH) from the adenohypophysis of the anterior pituitary. ACTH reaches the adrenal glands via the bloodstream, and activates the synthesis and release, in the adrenal cortex, of cortisol in humans and corticosterone (CORT) in rodents into the systemic circuit, where they travel back to the brain to bind to their cognate glucocorticoid (GR) and mineralocorticoid (MR) receptors. Of note, a large number of studies have shown sex differences in the secretion of CORT under both basal conditions as well as in response to precursors of CORT, and a variety of psychological and physiological stressors [11]. It is generally believed that these sex differences result from differential regulatory effects of circulating gonadal hormones on HPA-axis activity. While estrogens sensitize the HPA-axis response to stressors and increase basal HPA-axis activity, testosterone has the opposite effects [11]. In agreement with this, gonadectomy in female rats attenuated HPA axis activity, whereas removal of testes increased basal HPA-axis activity in males [12]. Sex differences in neuroplasticity as a result of chronic stress have also been reported [13], which is likely related to differential stress responsivity and susceptibility to diseases in males and females. These are important findings that may contribute to explaining why stress-related disorders, such as depression, post-traumatic stress disorder (PTSD), and anxiety, are more prevalent in females [14,15,16]. Better understanding of the sex-specific mechanisms underlying these disorders will allow for more targeted and effective treatment options.

Interestingly, despite the well described sex differences with regards to stress reactivity [17], the majority of rodent studies use male rodents only to avoid female estrous cycle being a confounding factor.

GR and MR are highly expressed in brain areas involved in cognitive function, such as the hippocampus and the frontal cortex. Excitatory feedback of GR binding from the amygdala, or inhibitory feedback from GR binding in the hippocampus, back into the paraventricular nucleus then enhance or ceases the release of CRF, completing the feedback loop. Of note, the expression and regulation of GR, as well as the interaction of sex steroids with GR, are sexually dimorphic [18], which needs to be taken into account when interpreting results.

At the cellular level, stress has been shown to modulate many aspects of neuronal function, including dendritic spine morphology [19,20,21,22,23], synaptic transmission [24,25], and the expression of trophic factors, such as brain-derived neurotrophic factor (BDNF) [26,27,28,29,30]. According to length, degree of severity, and developmental period of exposure to glucocorticoid stress hormones, these alterations in neuronal physiology may benefit processing in the short-term, but implicitly impart vulnerability to pathology in the long-term by forcing the system into a permanent state of adaptation and compensation. Prolonged exposure to CORT was shown to reduce neurogenesis [31,32] and promote apoptosis of GR-expressing hippocampal neurons [33], and to cause morphological abnormalities in the prefrontal cortex (PFC) [34].

Recently, with Thomas Insel, the head of the National Institute of Mental Health, as well as the recently established Research Domain Criteria (RDoC), suggesting that mental disorders be regarded as “brain disorders” [35], the role of stress within the brain becomes more pertinent, given that many psychiatric disorders—particularly those of affect, anxiety, and psychosis—can be triggered by and modulated by stress exposure. Psychiatric disorders, such as schizophrenia and major depression, are thought to be disorders of neurodevelopment with origins in early (pre- and postnatal) life. The mean age of onset of major psychiatric disorders is typically during adolescence or early adulthood [36,37], and it is well accepted that brain regions associated with cognition and stress reactivity, such as the hippocampus or prefrontal cortex, are still undergoing maturation during these time points, [38]. Thus, aberrant development during critical periods may underlie or impart susceptibility to psychiatric symptomatology, and hence, the need to understand the role of stress during different periods of development is a priority.

Given the complex nature of different types of stressors and the body’s differing responses to them, which are themselves compounded by a myriad of factors including genetics, environment (prenatal and postnatal) and sex, the study of stress and its fundamental effects in humans is extremely difficult. Rodent model is a tool that allows us to isolate and examine specific types of stressors at specific time points of development while minimizing genetic and environmental differences. This will allow us to uncover some important mechanisms of how stress can influence brain functions, which may offer knowledge important towards the treatment of human neurological disorders that are linked to stress. Keeping this in mind, the current review serves to evaluate the divergent behavioral effects of various types of stress, as modelled in rodents, during early, adolescent, and adult development. Due to the disparity in type, form, and length of stress treatments utilized between studies (see Figure 1), the discussion of physical, social, and pharmacological stress treatments will be separated to highlight important experimental outcomes of each form of stress when administered during different developmental periods. This review will particularly focus on long-term effects of stress on rodent behavior (such as cognitive function, depression- and anxiety-like behavior), as measured during adulthood. For the impact of stress on molecular changes, please refer to e.g., [10,39,40]. We will further outline the current understanding on sex differences in response to stress within each section of developmental time points, and emphasize their relevance in the respective context and the importance for future studies to differentiate and explore differences between the sexes, whereby important mechanistic insight may be gained. These effects of stress during early life, adolescence, and adulthood in males and females are summarized in Table 1. Given recent advancements in the accessibility of transgenic mouse lines, a brief discussion on how genetically modified rodent models can be utilized to delineate the behavioral effects of stress, with the GR and MR systems used as models.

## 2. Early Life Stress Models

Stress experienced during early life has a disruptive impact on the long-term outcome of an individual. During gestation, maternal stress is transmitted to the fetus via stress-related hormones, such as glucocorticoids. While it is beyond the scope of the current review, it is important to note that stress begins affecting the organism in utero. Despite being in the relatively cocooned environment of the womb, there is complex interplay between maternal genetics, offspring genetics, timing of stress, and the nature of stress, which together dictate the influence on offspring outcomes [87,88]. Prenatal stress can greatly impact upon the fetus and lead to permanent modification of the HPA axis, and stress responses later in life in a sex-dependent manner [87,89]. These results point to the significant role stress has, especially during critical phases of brain development, on the long-term function of the organism. Furthermore, the evidence underlines the fundamental role sex plays in mediating stress-induced effects, and advocates for the importance to include both sexes in rodent studies. After birth, the developing child faces many potential types of stressors, ranging from maternal deprivation or physical danger to psychosocial stressors, such as family conflict, neglectful parenting, or peer rejection. A large number of studies show that early stress becomes deeply embedded in the child’s neurobiology, with profound long-term effects on cognition, emotion, and behavior [90,91,92,93]. Stress exposure during early life stages has been linked to increased risk for neuropsychiatric disorders, such as depression, autism, or schizophrenia [94]. In order to understand the importance of stress in the etiology of these disorders, and develop new therapeutic targets, a variety of animal models are used. While models of adolescent or adult stress involve physical/psychological, social, or pharmacological stress (as discussed in the next sections), early stress mainly involves psychological stress, including maternal separation (MS), early deprivation or isolation (ED), or early handling (EH). MS involves either a single 24-h separation of the intact litter from the dam, or repeated separations for a shorter, but prolonged period of 3–6 h/day [95]. EH involves daily human handling of pups to separate them from the mother, and usually also from the littermates, for a short time period, typically 15–30 min/day [95,96], and ED comprises repeated (typically daily) human handling of pups and separation from the mother and littermates for a prolonged period of 1–6 h/day [95]. Please note that other paradigms, such as physical stress during early life or pharmacological manipulations are used as well, but to a lesser degree at this life stage. Hence, in this section, we will focus on the psychological paradigms and their translational relevance. Furthermore, it is well known that early stress may cause age-dependent, differential behavioral outcomes [97]. This section will focus on animal models of neonatal stress and the long-lasting effect on behavior in adulthood, summarizing and evaluating current knowledge in the research of early stress.

### Psychological Stress Models in Early Life

Studies by Seymour Levine have demonstrated that changes in the early postnatal environment (EH) can have lasting consequences for stress-responsiveness [98,99]. During this period, the presence of the dam is crucial for controlling activity of the HPA axis [100]. Alterations in HPA-axis activity can also be observed after a single separation for 24 h of male pups from the dam. Single 24 h MS in rats at postnatal day 3 (PD 3) resulted in increased basal CORT levels at 3 months of age, but this effect was not maintained into adulthood [101], while repeated MS for 3–6 h has been shown to affect HPA-axis responses into adulthood [102]. At the behavioral level, maternal separation has a profound effect on cognitive as well as emotional function. Male rats subjected to MS (3 h daily for the first 3 weeks) showed depressive-like behavior, as well as cognitive disruption accompanied by HPA-axis abnormalities in adulthood [41]. Supporting this data, Oomen et al. [42] also demonstrated spatial memory deficits in adult male rats after MS on PND3, however, this study further found that MS improved hippocampal neurogenesis and emotional learning under high stress conditions. These observations show that adverse early life events do not necessarily evolve into overall impaired hippocampal function later in life, but may provide resilience for optimal performance under stress in adulthood. Similarly, Champagne and colleagues [43] demonstrated that male adult low licking and grooming (LG) offspring displayed enhanced memory relative to high LG offspring, when tested in a hippocampal-dependent, contextual fear-conditioning paradigm. Hippocampal levels of glucocorticoid and mineralocorticoid receptors were reduced in low, compared with high LG offspring. Interestingly, low LG offspring, in contrast to those of high LG mothers, displayed significantly impaired long-term potentiation (LTP) under basal conditions, but surprisingly, a significantly enhanced LTP in response to high CORT in vitro. Furthermore, rats exposed to chronic stress in adulthood showed reduced immobility time in the forced swim test if previously exposed to neonatal handling, and this was accompanied by reduced neurochemical Na^+^, K^+^, and ATPase activity in the hippocampus, but increased activity in the amygdala [103]—thus, the amygdala, in particular, may adopt resilience to previous stress exposure. This aligns with human studies whereby van Harmelen et al. [93] showed that early emotional maltreatment is associated with enhanced amygdala activity. Similarly, Lupien and colleagues [104] demonstrated that the amygdala is larger in children (boys and girls) raised by mothers suffering from depression. Hence, in agreement with animal studies, human studies report that while hippocampal-dependent cognitive function is impaired, emotional responsiveness, largely modulated by the amygdala, seems to be enhanced in adult individuals with a history of negative early life experience [92,93,105]. However, it is important to note that the extent of maladaptation or adaptation in response to early stress depends on the intensity, duration and type of stressor, genetic background, as well as the time point of testing (adolescence, adulthood). Protocols of early stress, animals, animal strains, time window of stress application, as well as the testing period and tests used, differ between laboratories, which makes it hard to draw conclusions on the effect of stress, per se. The definition of adolescent period or adulthood differs between studies and animals, which makes it difficult to compare outcomes. Furthermore, due to our highly dynamic biological nature, changing and adapting in response to the environment stimuli occurs every second. It is not enough to define, for example, adulthood as one fixed biological time point. Suri et al. [44] demonstrated differential effects of early stress (MS 3 h daily from PND2 to PND14) in early adulthood versus middle-aged rats. While ES animals in young adulthood (2 months) exhibited improved spatial learning and enhanced anxiety, the same animals showed normal anxiety levels, but long-term spatial memory deficits later in life (15 months). These behavioral changes were accompanied by hippocampal gene expression changes, which varied between the two time points. Hence, the memory improvement described by other studies may only comprise a small window in early adulthood while it declines with age.

Of note, all early stress animal studies summarized in this section used male rodents only while the majority of human studies generally use both males and females, and sometimes treat both sexes as one group. It is well established that males and females have different responses to stress as outlined in the introduction, and it is important to understand these differences in order to understand their impact on stress-related disorders and to establish well-targeted treatments. More recent studies have begun to focus on the effects of early stress on both males and females. Although the scope of this review is postnatal stress, it is important to mention that differential response to stress begins as early as in the placenta. A review by Perez-Cerezales and colleagues [106] nicely summarizes human and animal studies examining sex-specific responses to environmental stressors during both the periconception period (caused by differences in sex chromosome dosage) and placental development (caused by both sex chromosomes and hormones). In terms of postnatal early stress, Liu and colleagues [45] showed that witnessing maternal trauma between PD 21 and PD 27 (dam of offspring was exposed to an aggressive male rat 3 times a day for 7 consecutive days, while the offspring PD 21–27 was placed within proximity) induced behavioral despair phenotype in forced-swim test at PD 60 in both male and female offspring rats, with greater effects in male rats. Forced-swim test, as introduced by Porsolt and colleagues [107], measures behavioral despair, a “depressive-like” phenotype in rodents that is sensitive to a wide range of antidepressant drugs. In this test, rodents are placed into a beaker full of water for 6 min, and their immobility time is recorded. It is believed that if mice “give up” earlier—meaning that they stop moving to try to escape—they are showing “depressive-like” symptoms of despair and helplessness. This forced-swim paradigm is sometimes also used as a stressor, which will be mentioned in section 3 (adult stress). Another study showed that male rats, following maternal separation during PD 1–7 for 15 min each day, exhibited increased preference for alcohol as compared to female rats during pubescent period and adulthood [47]. In our own laboratory, we previously subjected rats to early stress (MS) as well as adolescent stress (CORT treatment), singly or together, to delineate the effects of single stressors per se, and the combined effect of both stress insults on behavior, as well as molecular expression [46]. We showed that MS only decreased the expression of mature BDNF protein in male mice in the dorsal hippocampus, while BDNF was decreased in female mice in the ventral hippocampus in adulthood, showing sex-specific and region-specific effects of early stress on BDNF expression. BDNF plays an important role in hippocampal synaptic plasticity, neuronal migration, and protein synthesis-dependent long-term potentiation [108], and abnormal BDNF expression has been associated with depression and schizophrenia [109,110]. Albeit, no behavioral changes were seen after MS only, the combination of two stress insults (early and late) resulted in short-term memory impairment in males, while female rats showed anhedonia (inability to experience pleasure, a core symptom of major depressive disorder) as measured by the sucrose preference test, reflecting diverse manifestation of the same stressors between the sexes. Interestingly, a recent study in humans found that early adverse life events differentially affected male and female brains in terms of brain network architecture [111]. More sex-specific studies are needed to understand the sex-specific contribution of early life stress to the development of mental disorders, later in life.

## 3. Adolescent Stress Models

Adolescence is a transitional stage between childhood and adulthood with enormous physical and psychological changes, making it vulnerable to environmental insults, such as stress or drug abuse. It is well accepted that brain regions associated with cognition and stress reactivity, such as the hippocampus or prefrontal cortex, are still undergoing maturation during adolescence [38]. McCutcheon and Marinelli [112], for instance, showed that molecular mechanisms underlying long-term potentiation (LTP) differ in the adolescent period compared to both earlier life and adulthood. Hence, exposure to chronic stress during this critical period may result in persistent remodeling of brain structures critical for cognitive and emotional behaviors. A large number of reports link stressful life events, particularly during adolescence, with the onset of depression or other mental disorders, as well as drug use and cognitive impairment, suggesting it may be a triggering event [38,90,113]. Furthermore, it has been shown that periadolescent stress can affect stress responsiveness of the individual during and after pregnancy [114], thereby having the potential to affect stress adaptation in a transgenerational manner, through both behavioral and epigenetic modulation [115,116]. For example, maternal exposure to early life abuse has been linked to an increased risk in the offspring to various mental disorders, such as autism [117] and ADHD [118]. This is not surprising given evidence that the adolescent period is a time when the major endocrine system responsible for stress adaptation, the HPA axis, undergoes dynamic change in its regulation, both due to endocrine changes associated with puberty as well as in response to stressors [119]. For instance, while depression rates are roughly equal among prepubescent boys and girls, the rate increases dramatically once girls reach puberty, where, by age 15, females are twice as likely as males to have had an episode of depression [120]. Importantly, in rats and mice, the HPA axis is still maturing during adolescence and differential HPA-axis response to stress has been shown during adolescence compared to adulthood [121], suggesting, firstly, that puberty is a key phase of development across species, and secondly, that rodents can be a valid model to investigate the various intricacies surrounding this dynamic stage in relation to stress.

Even though adolescence was considered to be human specific, many claim to distinguish an adolescent period in rodents based on changes in developmental trajectories during the peripubertal period, in behavior, such as increased risk taking; neuronal development in the frontal-cortical and limbic brain regions; and changes in gonadal hormone levels [38], similar to the human definition of adolescence [122]. Puberty in rodents (as well as humans) is well defined, with physical changes in sexual organs as well as hormones [123]. In female rats, puberty starts with vaginal opening (VO), which occurs around PD 30 and ends with the first estrous cycle around PD 40, indicating full establishment of hormones for sexual reproduction (reviewed in [123] (Figure 2)). In male rats, according to Fernandez-Fernandez et al. [124], sexual organ maturation begins with the sign of balano-preputial separation (BPS) around PD 40, and ends with the presence of mature spermatozoa and the completion of spermatogenesis around PD 60 (reviewed in [125] (Figure 2)). Similar biological processes defining puberty were observed in mice. While female mice show sexual maturation between PD 25 and 35, male mice show a later onset and longer period of puberty (PD 27–40) (reviewed in [125]).

In this section, we will briefly summarize the most recent insights into the models of adolescent stress. We will particularly focus on chronic stress during adolescence, and its long-lasting effects on cognitive performance. While early stress models mainly involve psychological stress, as described in the previous section, stressors used for adolescent animal models predominantly comprise physical, social, and pharmacological stressors, as outlined below. Once again, the majority of the studies used only male rodents, and only a few studies discriminated between males and females, which will be emphasized at the end of each section.

### 3.1. Physical and Social Stress during Adolescence

Physical stressors used in animal research to date mainly include noise stress, foot shock, or circadian rhythm changes, while social stress is mimicked by social isolation, litter shifting, introducing new cage mates, or exposure to novel environments. In contrast to physical stress, psychological stress paradigms attempt to isolate the neurobiological aspect associated with fear and anxiety, rather than inducing direct physical challenges, such as discomfort or pain to elicit a stress response. One method is to expose rodents to predatory odors. Some researchers have gone further to expose rodents to actual predators, such as a cat [126,127]. Compared to other methods, researchers have argued that predator exposure is a more ethologically relevant inducer of stress that mimics emotional states related to fear and post-traumatic stress disorder in humans [128,129].

Human studies showed that multiple unpredictable stressors were more predictive of psychiatric disorders than a single adversity alone [130]. Therefore, many studies apply a variety of unpredicted stressors in the animal research to strengthen the predictive validity and represent constructs that are more likely to mimic the human condition. Chronic unpredictable stress (CUS) can entail a range of either physical or social stressors, applied randomly, to inhibit habituation. A research group using a combination of physical and social stressors from PD 30–70 in male rats showed impaired learning behavior in these rats during adulthood when tested in the radial water maze, while working and spatial memory remained intact [48]. To discern potentially different effects between social and physical stressors, Isgor and colleagues [50] exposed rats to a variety of physical stressors (restraint, loud noise, cold exposure) or social stressors (isolation, novel environment, litter shifting) over four weeks from PD 26 to 58, and compared the effects of social CUS versus physical CUS on morphological and behavioral changes in these rodents. They demonstrated that three weeks after the last stressor, social-CUS animals did not show hippocampal volume changes or spatial memory deficits as opposed to the physical-CUS group. This indicates that the type of stressor is critical to cognitive performance later in life, and social stress according to this observation seems to affect spatial memory in rodents to a lesser degree, as compared to physical stressors. However, in a variation of chronic social stress, where new cage mates were introduced twice a week for 7 weeks beginning at day 32, impairments in spatial memory, as shown in the Y-maze and working memory in the Morris water maze, but not in an object or social recognition task, were found when tested one year after stress exposure [55], suggesting long-term effects of chronic social stress on aspects of memory.

Deficits in spatial performance were also shown in male and female rats after social instability stress (1 h isolation and change of cage mates daily from PD 30 to 45) [56,131]. Morrissey and colleagues [58] showed reduced contextual fear conditioning in rats exposed to social instability during adolescence, but not during adulthood. This suggests that the type of social stress and the timing plays a crucial role in determining whether it will affect a particular type of memory. In support of the study by Isgor et al. [50], claiming that physical CUS has a profound effect on spatial memory and hippocampal volume as mentioned above, exposure of rodents to physical variable stress resulted in reduced hippocampal GR receptor expression, as well as increased hippocampal volumes [51]. These morphological changes were associated with spatial memory deficits in adulthood [51]. Oztan et al. [52] demonstrated reduced social interaction and forced swim immobility, suggesting that adolescent exposure to physical CUS not only results in cognitive impairment, but also in depression-like behavior. Notably, the same stressors given in adulthood did not elicit such persistent morphological and behavioral changes (reviewed in [51]), suggesting the adolescent period to be specifically vulnerable to CUS.

Unfortunately, not many studies have looked at sex differences with regards to the effects of chronic adolescent physical and social stress on adult behavior. One study by Pyter and colleagues [49] investigated the effect of chronic stress on neuroimmune effects during adulthood in male and female rats. The neuroimmune system is highly interconnected with the endocrine stress system, and both are suggested to contribute to psychiatric disorders, such as major depression [132]. In this study, male and female rats underwent a chronic adolescent stress paradigm, in which experimental rats were exposed to randomized episodes of restraint stress and social defeat by same-sex aggressors. When challenged with the endotoxin liposaccharide (LPS) intraperitoneally in adulthood, male stress-exposed rats displayed exaggerated induction of the pro-inflammatory cytokines IL-1β and TNF-α. Interestingly, females did not display a similar inflammatory response [49]. This study suggests males to be more vulnerable to adolescent stress than females, in terms of the immune response in the brain. A recent study exposed adolescent male and female C57BL/6J mice to chronic variable social stress (CVSS; repeated cycles of social isolation + social reorganization) or control conditions from postnatal days (PD 25–59) [54]. Anxiety-like behavior was measured in the elevated plus-maze at PD 61–65, and synaptic transmission in the prefrontal cortex (PFC) and nucleus accumbens (NAC)—brain regions that are implicated in anxiety and addiction—were assessed at PD 64–80. While both male and female mice developed anxiety-like behavior, stress decreased the amplitude of spontaneous excitatory postsynaptic currents in the PFC only in male mice, while these were decreased in NAC specifically in female mice. This study emphasizes sex-specific and brain region-specific differences in response to chronic adolescent stress. More studies are needed to better understand the involvement of sex hormones in response to stress on brain development in males and females.

Taken together, it appears that physical CUS, specifically during adolescence, but not during adulthood, has a more profound effect on spatial memory compared to social CUS, as measured in male rodents. Social CUS, however, can elicit memory impairment as well as depression-like behavior, dependent on the duration and type of CUS. The aforementioned sex-specific and brain region-specific differences in response to chronic adolescent stress further emphasize the need for more studies delineating differential effects of stress in males and females.

### 3.2. Pharmacological Stress During Adolescence

The adrenal steroid hormone glucocorticoid (cortisol in humans and CORT in rodents) is a major stress hormone that has a range of physiological effects. Prolonged elevation of glucocorticoids, however, has been shown to be damaging to the brain, particularly in the hippocampus [33,83,84]. Administration of glucocorticoids has been used as a pharmacological model of stress, however, it is important to keep in mind that elevating stress-hormone levels does not represent a natural stress response, and is therefore difficult to translate to human conditions. Moreover, the administration of CORT (drinking water, pellets, injections), its form (corticosterone or corticosterone 21-hemisuccinate) and the dosage, plays an important role, and slight changes in these factors may elicit different responses. The highest densities of glucocorticoid receptors are found in the hippocampus, frontal cortex, and the amygdala, structures involved in cognitive function [133], as well as regulation of the stress axis [134]. While the hippocampus is believed to mediate spatial memory [135], the prefrontal cortex is involved in executive function, such as goal-directed behavior or attention. Interestingly, in the literature of animal research, adolescent CORT administration mainly affects hippocampal morphology, but does not appear to elicit any behavioral phenotype. One particular study showed that while chronic adolescent stress induced by restraint stress (6 h daily for 3 weeks) impaired spatial short-term memory, no such effect was seen after CORT administration in drinking water (40 mg/L) during the same period; however, both stressors reduced dendrite arborization in the hippocampus [53]. This highlights a more profound effect of chronic restraint stress during adolescence on cognitive function, as compared to CORT treatment alone, and suggests involvement of other stress hormones, such as adrenaline. In line with this study, our own data did not show any deficits in spatial short-term memory in rats after administering CORT in drinking water (50 mg/L) for 3 weeks (8–10 weeks of age), albeit that rats were given a two-week gap between the end of CORT treatment and behavioral testing [46]. In another study by our group, after administering CORT (50 mg/L) from week 6–9 in mice, we uncovered a male-specific disruption to sensorimotor gating as measured by pre-pulse inhibition (PPI) [59]. Disruptions in sensorimotor gating are frequently reported in schizophrenia patients [136]. Our study suggests that males are more susceptible to develop PPI deficits when exposed to high stress-hormone levels during adolescence as compared to females.

Overall, CORT treatment during adolescence appears to have either no effects or induce subtle molecular changes that do not transpire to behavioral abnormalities, but may be a triggering event when combined with an additional environmental or genetic insult [46,59,137,138]. For instance, we found that male, but not female mice that are heterozygous for the *BDNF* gene, when given adolescent CORT treatment, developed short-term spatial memory deficits in adulthood, as shown by the Y-maze [137]. Furthermore, sex-specific deficits were also uncovered when mice that were exposed to maternal separation (PND 2–14, 3 h/day) were given CORT treatment during young adulthood [138]. This “two-hit” paradigm resulted in a male-specific deficit in short-term spatial memory as shown by the Y-maze. Interestingly, females exposed to the “two-hit” treatment displayed anhedonia phenotype in the sucrose-preference test. These behavioral phenotypes were matched with reductions in mBDNF protein level, specifically in the dorsal hippocampus in the male two-hit mice, while mBDNF level was specifically reduced in the ventral hippocampus in the female two-hit mice. These lines of evidence suggest that increased stress hormone during the adolescent phase can induce a vulnerability seeded by earlier stress experiences and result in sex-specific pathologies. While males appear to develop memory deficits when exposed to adolescent CORT treatment as a second hit, females tend to develop depression-like behavior.

Adolescence is one of the most dynamic periods of brain and body development, second only to infancy. However, unlike infancy, an adolescent is much more exposed to the myriad of stressors. The available evidence suggest that this period is acutely sensitive to stress-induced changes that can pervade into adulthood. Given sexual maturation occurs during this time, it is unsurprising that sexually distinct patterns of stress adaptation also occur, leading to different outcomes in response to stress. A better understanding of the mechanisms underlying the interplay of environment, genetic, and developmental factors during this crux phase may offer novel opportunities to prevent or treat stress-related disorders. In terms of face validity of the above-described models of adolescent stress, we observed that the majority of studies investigating the effect of adolescent stress focus on behavioral paradigms relevant to mental illnesses, rather than cognitive function, per se [57,139,140]. Similarly, human studies examining the effect of adolescent stress tend also to focus on mental illness rather than cognition [141]. Irrespectively, those studies focusing on cognition in humans tend to show cognitive abnormalities in tasks that are mainly mediated by the PFC [90], whereas the majority of rodent studies focus on the hippocampus-dependent memories, such as spatial short-term or long-term memories, as described above. Hence, the face validity of adolescent stress paradigms has to be strengthened by an increased focus on PFC-related tasks in animal models, such as, for example, the 5-choice serial reaction time task, using touch-screen apparatus, to assess similar types of PFC-dependent cognitive function, which is seen to be affected in humans.

## 4. Adult Stress Models

The association between severe or prolonged stress and subsequent development of psychiatric disorders is encapsulated by acute stress disorder (ASD) and post-traumatic stress disorder (PTSD), the diagnoses of which are both dependent on the experience of a traumatic event. However, a plethora of psychiatric disorders, such as depression and anxiety, are also intimately linked with stressful experiences, not only during developmental stages, but also during adulthood. Despite the well-established observation that PTSD as well as depression are more common in women compared to men [14,15,16], once again, not many studies have examined sex differences within this context.

### 4.1. Physical and Social Stress in Adulthood

With regards to physical stress, Crema and colleagues [61] showed that chronic restraint stress, 5 days a week for 40 days starting at PD 60, resulted in anxiety-like behavior in male Wistar rats. In male mice, a similar paradigm of 2 h of restraint stress a day for 21 days, starting at PD 38 as well as physical CUS, resulted in body weight loss and a depression-like phenotype measured via the tail-suspension test [63]. Other versions of chronic restraint stress have subjected rats to a daily session of 12 h duration for 2 days, which suppressed immunity by inducing lymphocyte apoptosis [62]. Interestingly, this echoes clinical findings where, following traumatic stress in humans, there is a reduced immune response [142]. Chronic immobilization stress for 2 h a day over 21 days in 2-month-old male rats resulted in a deficit in recall memory in the Morris water maze [64]. Adult male Lister rats exposed to a variety of physical stressors (physical CUS) over 5–9 weeks developed an anhedonic phenotype, as measured by the sucrose preference test [65]. A similar 4 week paradigm also resulted in anhedonic behavior, and increased submissive behavior in the residential intruder test, but an anxiolytic phenotype in the elevated-plus maze [66]. Even a relatively short 10-day paradigm was enough to elicit anxiety-like behavior in male rats, as shown in the defensive burying test, despite no alterations in the phenotypes in the elevated plus maze and the light/dark box [67]. In 10-week-old male mice, a 54-day paradigm of twice-a-day exposure to various stressors, including physical and social stressors, resulted in depressive-like, anxiety, and submissive phenotypes [60]. Another study compared 4 weeks of physical CUS (such as restraint in a plastic tube, cage tilting, placement in an empty cage with no nesting, placement in crowded cages, lights on for a short period of time during the dark phase, and white noise) and chronic restraint stress of the same duration in adult mice, and found that whilst both regimes produced anxiety phenotypes, only the CUS was able to elicit depressive-like behaviors [68]. It should be noted that strain differences exist as shown by a 3-week protocol which resulted in different phenotypes in two strains of inbred mice, with the C57BL/6 but not ICR mice developing a depression-related phenotype [143]. The majority of the abovementioned studies mainly report depression-like and anxiety-like phenotype in response to physical stress, and CUS appears to have a more profound effect on depression-like behavior compared to a single chronic stressor. Not many studies reported cognitive impairment after adult physical stress except one, which showed a deficit in recall memory after 21 days of immobilization stress in male rats [64].

In terms of social stress, in male rats, social defeat stress (introducing an intruder into the cage) induced depression-like behavioral phenotypes, impaired memory, and increased oxidative stress and inflammation [72]. In agreement with this study, Martin et al. [73] showed that chronic defeat stress for 10 consecutive days starting at PD 48 induced long-lasting anxious-like phenotype in the open field and episodic memory deficits in the novel object recognition test in male C57BL/6J mice. Another study showed that social defeat stress for 21 days in adult mice resulted in disrupted spatial memory accompanied by reduced complexity of apical dendrites of CA3 neurons [74]. Interestingly, a study has found that social defeat in patients predicted diagnosis of PTSD and course of disease remission [144]. Taken together, while the majority of the studies using physical stressors report depression-like or anxiety-like behavior in rodents, social stress appears to affect hippocampus-dependent cognitive function as well as induce depression-like behavior.

Exposure of rodents to cat urine, ferret urine, or trimethylthiazoline (TMT), a synthetic compound distilled from fox feces, can elicit various stress responses such as freezing, avoidance, and increases in stress hormones [145,146]. Areas such as the paraventricular nucleus of the hypothalamus (PVN), medial amygdala (MeA), and dorsal periaqueductal gray (PDAG) are intimately involved in the response to chemosignals, inducing autonomic, endocrine, and behavioral responses [147,148]. Predator exposure has been found to elicit long-lasting anxiogenic behavioral [149] and pathological neuroadaptations in the expression of synaptophysin and cannabinoid systems, both of which are involved in PTSD [150,151]. A recent study found that exposure to TMT induced impairments in memory retrieval in the radial arm maze in male mice [152].

Disruptions to the normal circadian rhythm can upset both the physical and psychological wellbeing of an individual [153]. Likewise, in mice, it has been shown that exposure to constant light for three weeks caused depressive-like and anxiety-like behaviors but decreased CORT [75]. Other studies found no changes in memory or anxiety parameters after 3–4 weeks of constant light exposure [77]. Craig and colleagues used, in Long Evans rats, a chronic phase shifting protocol. Light-off time was brought forward for 3 h each day for 6 days before 10 days of re-entrainment light-off at 22:30. This 16-day protocol was considered one session. Rats exposed to 4 consecutive sessions (at PD 64) developed hippocampal memory deficits in the Morris water maze test [78]. In the diurnal Nile grass rat, introduction of light in the dark phase for 3 weeks induced a range of phenotypes, including anhedonia, as shown by the sucrose preference test, increased immobility in the forced-swim test (FST), and impaired learning and memory in the Barnes maze. Concurrently, night time light reduced dendritic length in the DG and CA1 [76]. Adaptation by the animals to the change in day–night pattern is a major drawback, as no long-term study can be conducted. For this reason, most studies use this as a part of an unpredictable stress paradigm. Interestingly, changes in circadian rhythm seem to have a profound effect on depression-like behavior, as well as memory performance, as shown by several studies, which were also reported after social defeat stress.

While the aforementioned data is based on male rodents only, more recent studies began to focus on differential response to stressors in males and females. Viera et al. [69] recently compared the effects of repeated restraint stress over 10 days versus chronic variable stress over 10 days on 60-day-old (adult) male and female rats. This study showed that irrespective of stress type males were more vulnerable to somatic effects of chronic stressors, while females appeared to be more susceptible to neuroendocrine and behavioral changes, and developed anxiety-like behavior. Indeed, depression, as well as PTSD, have a higher prevalence in females [14,15,16], and may result from a differential response to stress in males and females. Bowman and colleagues [70] investigated the impact of the duration of restraint stress on short-term spatial memory performance in female rats. Twenty-one days (6 h per day) of chronic restraint stress at PD 70 enhanced female 8-arm radial arm maze (RAM) performance (spatial memory) in female rats, while 28 days neither enhanced nor impaired performance [70]. This pattern of results is different from male rats, in which prolonged exposure to stress changes from enhanced spatial memory at 14 days of adult stress to maladaptive (RAM performance was impaired following 21 days of restraint) [71]. These results are consistent with the data from male studies as outlined at the beginning of the section, showing that physical stress during adulthood for longer than three weeks has maladaptive effects.

### 4.2. Pharmacological Stress in Adulthood

Administration of glucocorticoid in rodents has been found to reduce hippocampal neurogenesis [31,32,80], increase apoptosis [33,81,83,84,154], and reduce volume and dendritic density of the prefrontal cortex [34,155]. Behaviorally, chronic oral CORT administration (100 ug/mL) to adult mice resulted in depressive-like behavior in the FST [79]. In rats, chronic injection of the more modest 20 mg/kg dose of CORT elicited depressive-like behavior in the FST, decreased body weight and, in a subset of high anxiety rats, increased anxiety-like behavior in the open field and elevated plus maze test [82]. However, the relationship between levels of circulating glucocorticoids and pathological outcomes is complex. For example, lower baseline cortisol levels have been found in patients suffering PTSD as a result of acute respiratory disease syndrome [156]. In another study, low dose cortisol (10 mg/day) was administered daily for 1 month to three patients with chronic PTSD and resulted in significant improvements [157]. An acute intravenous injection of a high dose of hydrocortisone (100–140 mg) to patients with acute stress symptoms within the first 6 h of a traumatic event was able to attenuate symptoms of the acute stress, as well as the subsequent PTSD in the 25 patients [158].

Similar findings have been reported in rodents [159]. This seemingly inconsistent effect may be due to the memory mediating effect of glucocorticoids—initial increases in glucocorticoid levels in response to stress can act to obstruct long-term memory formation of the traumatic event [160]. Hence, the use of exogenous glucocorticoid alone in stable dosages may only represent a single dimensional aspect of stress—that is, the effects of elevated glucocorticoid—and may miss the dynamic flux of the organism as it attempts to regain (or fail to, in pathological cases) homeostasis. It is also important to keep in mind not only the difference between the species (mice, rats), but also between strains, have to be taken into account when interpreting the effects of CORT. Chronic CORT administration (pellets of 20 mg CORT) to C56BL/6 mouse substrains J and N at 12 weeks of age for 3 weeks showed differential behavioral and molecular responses [85]. While C56BL/6N mice developed depression-like behavior, C56BL/6J mice remained resistant to adolescent CORT exposure [85]. While many studies purely focus on male mice, Mekiri and colleagues [86] used only female C57BL/6J mice to investigate the effect of chronic CORT administration to model an anxio-depression-like phenotype. They showed that 4 weeks of CORT exposure at 35 g/L in the drinking water enhanced the emotionality score of female mice, but with a very small effect size. Tests of longer treatment duration, however, failed to potentiate the behavioral effects of CORT. CORT had no effect on cell proliferation, survival, or neuronal maturation in the dentate gyrus of the hippocampus in this female model, suggesting other stress mechanisms play a more profound role in anxiety-like behavior, rather than stress-hormone levels, per se, in females. Taken together, although inconsistencies exist between studies, the majority of the literature report that chronic CORT administration during adulthood induces depression-like and anxiety-like behavior, decreased neurogenesis, and dendritic arborization (mainly in cortex and hippocampus) in male rodents. Female rodents, on the other hand, appear to be protected against high CORT levels during adulthood, and did not show any molecular differences, and even exhibited greater emotionality scores after four weeks of CORT exposure. The available evidence strongly advocates for the need to include both male and female animals in modelling stress-induced pathologies.

## 5. Genetically Modified Animal Models and the Study of Stress

Recent advances in mouse genetics within the past two decades has allowed for the generation of multiple mouse lines of relevance to the study of stress. The HPA axis is comprised of many regulatory elements, which, when disrupted, may result in abnormal stress reactivity. Examples include a polymorphism (5-HTTLPR) within the promoter region of the serotonin transporter gene, where a short allele, with a 43 bp deletion, results in hypersensitivity of the HPA axis [161,162,163]. Another example is the single nucleotide polymorphism (rs53576) in the oxytocin receptor gene, which has been shown to modulate HPA-axis stress responses [164]. However, there is commonality in their function to control circulating stress-steroid hormones from acting on their principal binding sites—the glucocorticoid (GR) and mineralocorticoid (MR) receptors [165,166]. Therefore, the study of GR and MR signaling reflects two of the most downstream events involved in the behavioral adaptation to stress. Early pharmacological studies implicated GR and MR receptor function in many behavioral processes, highlighting their functional significance. However, as previously noted, these studies lacked temporal, regional, or cell-type specificity for either receptor, which affected the overall utility of these models [167]. Given the generation of at least eight distinct GR and three MR mutant mouse lines [168], which vary in their genetic construct to overexpress, underexpress, knock-out (KO), and determine mechanisms of signaling, these receptor systems represent model systems to outline how various genetic manipulations can be used to deduce the mechanisms and pathways involved in the behavioral response to stress. In this section, these genetically modified GR and MR animal models will be briefly discussed where phenotyped, to outline how genetically modified animal models may be used to further tease out the mechanisms involved in behavioral adaption to stress. 

### 5.1. Early Insight from Non-Specific Loss- and Gain-of-Function Models

Countering claims that pharmacological models were too non-specific [167], the generation of GR and MR KO mice presented a leap forward in receptor specificity. However, GR^−/−^ and MR^−/−^ KO mice often die within hours or days following birth by means of respiratory failure [169] and renal Na^+^ loss [170], respectively. It is possible to extend the lifespan of MR^−/−^ KO mice by administration of exogenous NaCl; a strategy that has been used to implicate a putative role of MRs, and not GRs, in maintaining granule cell neurogenesis within the hippocampus of adult mice [171]. However, while exogenous salt treatment may extend the lifespan of MR^−/−^ KO mice into adulthood [171,172], neither MR^−/−^ nor MR^−/+^ mice appear to have been extensively behaviorally phenotyped to date, and it remains unclear whether MR-mediated granule cell neurogenesis within the hippocampus of salt-treated MR^−/−^ KO mice alters behavioral output. Further, despite reports that 10% of GR^−/−^ mice survive to adulthood [173], only GR^+/−^ mice have been behaviorally phenotyped; presumably because it is unknown if the severity of other developmental anomalies [169] may otherwise affect behavior. Speculation aside, GR heterozygote mice have been shown to have normal CORT levels at baseline, but show extended CORT elevations 40 and 60 min following restraint stress, while an exaggerated CORT response was also observed following challenge with dexamethasone (DEX) and DEX + CRF [174]. Behaviorally, no evidence of anxiety-related behavior (as evaluated using the elevated O-maze, light–dark box, and fear conditioning) was observed amongst GR^+/−^ mice in this study, however, on a shuttle-box test of learned helplessness, GR^+/−^ mice were shown to have increased escape latencies and failures suggesting a depressive-like phenotype; a finding which coincided with decreased BDNF protein expression within the hippocampus of GR^+/−^ mice [174]. Interestingly, within this same study, the authors also behaviorally phenotyped a GR overexpressing transgenic mouse line that had been genetically modified to carry two extra copies of the GR gene by means of a yeast artificial chromosome [175], allowing for the direct comparison of the behavioral effects of GR down- and upregulation. These mice, termed GR^YGR^ transgenic mice, were shown to have a phenotype opposite to that of GR^+/−^ mice, by showing decreases in HPA-axis reactivity following restraint stress and challenge with DEX, a resilient phenotype on the shuttle-box learned-helplessness paradigm, and an increase in hippocampal BDNF relative to controls [174]. It is worth noting, however, that the expression levels of GR^+/−^ and GR^YGR^ mice are not proportionate. In this study, GR heterozygote mice showed a downregulation of GR mRNA to 33% of littermate controls, while GR^YGR^ mice showed upregulation of GR mRNA to 219% of littermate controls [174], which suggests that, based on the relative difference in gene expression between the two models, disruptions to GR gene expression has the ability to shape behavioral processes. Lastly, an alternative loss of function model, where animals carry an inverted antisense GR cDNA allele under the control of a putatively neuronal-specific neurofilament promoter [176,177], also showed evidence of a resilient phenotype by showing reduced immobility and an increased latency to first immobility on the FST [176]. That said, unlike GR^+/−^ mice, these antisense GR (AGR) mice showed a significant increase on number of arm entries and time exploring the open arm on the elevated plus-maze, and a selective deficit on a social-recognition paradigm compared to controls, which were normalized by the antidepressant moclobemide [176]. Despite trying to control expression of GR specifically within the brain, these AGR mice also reportedly suffer from alterations in GR expression within peripheral tissue [168], making it less specific than the more modern, and regionally-specific, conditional knockout strains.

### 5.2. Insight from Brain-Specific and Forebrain-Restricted Depletion of GR and MR

The first study to generate conditional Cre/LoxP GR KO mice, using nestin (Nes) to direct deletion of GR within the nervous system, reported dysregulated basal CORT levels, reduced ACTH levels, and overexpression of CRF within the PVN [178]. Promisingly, these GR^NesCre^ mice show no major morphological anomalies within the adrenal glands, which are otherwise severely affected in GR^−/−^ KO mice [169], but do show evidence of adrenal sensitivity on ACTH stimulation tests [178]. The behavioral phenotype of these GR^NesCre^ mice was associated with an anxiolytic-like phenotype, whereby mutant mice showed a reduced latency to enter the “light” environment, and spent more time exploring it, on the light–dark box test, which was complemented by longer exploration times of the open section of the elevated zero-maze relative to controls [178]. Further to this, GR^NesCre^ mice also showed a resilient phenotype on the forced-swim test (FST), where there was no significant difference in immobility compared to controls on day one of testing, but when repeated on day two, GR^NesCre^ mice swam for significantly longer than controls. While this test has been classically used as a measure of learned helplessness, of relevance to affective disorders, it has also been suggested that this FST phenotype of GR^NesCre^ mice may represent a cognitive deficit whereby mutants are unable to learn that attempts to escape are futile [168]. In a separate study, and of relevance to addiction, GR^NesCre^ mice were shown to have a “flattened” dose–response function to, but showed intact acquisition of, intravenous cocaine self-administration, while behavioral sensitization to cocaine was completely suppressed [179]. Molecularly, GR^NesCre^ mice show downregulated expression of synapsin isoforms Ia/Ib, but not IIa or IIIa, and the transcription factor Egr-1 within the hippocampus [180]. Linking this to behavioral function, wildtype mice, which undergo contextual fear conditioning followed by intra-hippocampal infusion of CORT, selectively show upregulated synapsin-Ia/Ib expression and enhanced contextual fear, which suggests a putative signaling pathway, comprising GR-Egr1-MAPK-Syn-Ia/IB, related to the processing of stress-related memory [180,181]. Based on this result, it would be of interest to examine whether this same signaling pathway within GR^NesCre^ mice plays a role in the renewal of contextual fear following extinction learning, and may be an avenue of further research.

A step-up in specificity, forebrain-specific ablation of GR and MR signaling has implicated the role of these receptors in limbic function. Mice with a forebrain-specific deletion of GRs, which were also developed using the Cre/LoxP system, are termed GR^CaMKCre^ for being under the control of calmodulin kinase, and show evidence of HPA-axis hyperactivity, increased behavioral despair, and depressive-like behavior on the sucrose preference test, FST, and tail-suspension test (TST) [182]. While treatment with the antidepressant imipramine reversed the FST and TST phenotype [182], a separate study assessing both males and females replicated the depressive-like phenotype of male GR^CaMKCre^ mice, but failed to find evidence of HPA-axis hyperactivity nor depressive-like behavior amongst female GR^CaMKCre^ mice [183]. Similarly, when crossed onto a new foundation line with a pure C57BL/6 background, the deletion of forebrain GRs once more failed to induce this male-specific phenotype [184]. While these results highlight sex differences and the fact that antidepressant treatment appears to act independently of forebrain GR expression, these contradictory behavioral results suggest the need for further study before these data can be extrapolated to clinical cases. On the other hand, MR^CaMKCre^ mice also show no disruption in basal nor stress-induced HPA-axis reactivity relative to controls, and show no evidence of an anxiety-related phenotype on the open field, elevated O-maze, and light–dark box tests [185]. That said, deficits in the acquisition and reversal phases of the Morris water maze, specifically during the early phase of testing, were observed amongst MR^CaMKCre^ mice, while on the radial arm maze, the working memory of MR^CaMKCre^ mice was also selectively disrupted. Within the hippocampus, MR^CaMKCre^ mice showed altered mossy fiber projections and significantly increased GR expression within the Cornu Ammonis (CA) field [185]. A more recent study also found that MR^CaMKCre^ mice also showed delayed learning on the circular hole board (CHB) test, and while 5–10 min of acute restraint stress disrupted task learning amongst the control group, the MR^CaMKCre^ mice remained unaffected; a result which suggests a role for MRs in the stress-induced formation of hippocampus-dependent spatial memory [186]. It is worth noting that this study utilized a male-only sample. In a separate study by the same group, only female MR^CaMKCre^ mice were sampled, and a similar CHB deficit was observed, but only when mice were in proestrus and estrus. Further to this, stress-susceptibility on CHB performance was also observed but was once more specific to MR^CaMKCre^ female mice in estrus [187]. Cumulatively, these studies show a selective role for MRs in the early formation of spatial memory traces, and that stress-susceptibility may be dependent on other endocrine systems, highlighting the need to include sex as a factor in the investigation of complex interactions in the behavioral modelling of stress.

### 5.3. Insight from Deficient DNA Binding in GR^Dim^ Mice

As GRs exert their action by DNA binding both directly, via glucocorticoid response elements (GRE), and indirectly, via interactions with transcription factors, the behavioral dissection of these two pathways is, therefore, a necessary but challenging task. One model, however, involving the insertion of a point mutation termed A458T within the D loop, a region of 5 amino acids within the DNA-binding domain of GR located in exon 4 [188], which affects dimerization, DNA binding, and subsequently, the transactivation of GRE-containing promoters [189]. Mice carrying this genetic variant, termed GR dimerization-deficient or GR^Dim^ mice [188], are therefore deficient in dimer-required DNA-mediated GR function, but not monomer-mediated protein–protein interactions with other transcription factors [190], allowing for pathway specificity when interrogating the behavioral effects of GR function. Correspondingly, GR^Dim/Dim^ mice have been shown to have unchanged CRF expression within the PVN and median eminence, suggesting that GR control of CRF may be dimer independent, while POMC expression was increased within the anterior pituitary [188]. ACTH immunoreactivity was increased 2.2-fold in the anterior pituitary as expected, however, serum ACTH was found to be unchanged, which the authors suggested could be the result of a dimerization-independent secretion mechanism. However, radioimmunoassay did reveal increased basal CORT levels [188], which remain elevated relative to controls following one minute of swim stress after 30 and 90 min [190]. Behaviorally, significantly longer swim distances and latencies to locate the platform than controls on the water-maze have been observed, suggesting impaired spatial memory, but unlike other models, no evidence of an anxiety-related phenotype on the light–dark box and open field test has been observed [190]. Given that the dorsal hippocampus is putatively a critical mediator of spatial memory [191], while the ventral hippocampus is believed to have a more prominent role in anxiety-related behavior [192], it is interesting to note that GR expression is approximately two-fold higher in the dorsal hippocampus relative to the ventral hippocampus in both rat and mouse [193,194]. In this regard, it is possible that in GR^Dim/Dim^ mice, the dorsal hippocampus is more severely affected by the disruption of GR dimerization and DNA-binding processes than the ventral hippocampus, resulting in memory impairment rather than an anxiety-related phenotype. Complementary to this interpretation, the behavioral phenotype of GR^Dim/Dim^ mice may also suggest that spatial memory deficits are under the control of GR dimerization, while anxiety-related behavior is mediated by GR monomer action [195], providing distinct regions and signaling pathways by which glucocorticoids may regulate behavioral and cognitive processes. It would be interesting to further evaluate whether there are sex-specific differences with regards to GR dimerization and DNA-binding processes.

## 6. Conclusions

Here, we compared and summarized the most recent evidence of rodent stress models used in research to better understand the mechanism underlying chronic stress. Due to an enormous amount of data in stress research, including numerous types of stress within different species and time points of stress application, we organized this review into three major time points of chronic stress application: (1) postnatal early life, (2) adolescence, and (3) adulthood. We further only focused on how this chronic stress affects mainly behavior, but also molecular composition in adulthood. For each section, we classified the type of stress into physical, social, and pharmacological stress to better distinguish and interpret the behavioral and molecular results.

Despite the mentioned variability and inconsistency between studies in terms of type/duration of stress, window of stress application, and type of test and timing of assessment, we could see major parallels of early, adolescent, and adult stress on behavioral outcome. Psychological stress during early postnatal life can have a long-lasting detrimental impact on cognitive function, but can also create resilience, particularly to fear-potentiated memory, depending on the exact stress paradigm and timing of the stressor. It may, indeed, explain the notion that early stress could in fact prepare the organism to function better under high stress later in life, when the individual possesses a certain predisposition to stress reactivity, dictated by early experience. Furthermore, with only a few sex-specific studies, males seem to be more susceptible to early life stress compared to females. No studies of stress application during adolescence showed any beneficial effects on behavior. These studies mainly report abnormal depression-like or cognitive behavior with molecular abnormalities mainly in the hippocampus or the prefrontal cortex. A few studies showed that physical stress has a higher impact on the stress response compared to social stress. Sex and brain specific abnormalities were reported after adolescent physical and social stress, with males tending to fare worse in terms of hippocampal-dependent memory outcomes. With regards to CORT application during adolescence, our own laboratory showed a male-specific disruption in sensory gating as measured by PPI, suggesting that CORT has a greater effect in males compared to females, and in a two-hit paradigm, maternal separation followed by adolescent CORT treatment caused spatial memory deficits in males, but anhedonia in females, showing sex-specific effects of adolescent CORT exposure. Interestingly, according to the studies on adolescent/adult pharmacological stress and GR receptor conditional KO mice, females were not affected or less affected than males in terms of developing a behavioral despair phenotype. This suggests that other stress mechanisms must be involved in females besides the glucocorticoid–GR signaling pathway, while males seem to be more affected by alterations to glucocorticoid signaling.

Notwithstanding the current valuable advances in stress research using animal models, investigators should be more mindful of which type of rodent, including strain, age and sex, which type of stressor, and what time points/duration of stress application/behavioral or molecular assessment to use in order to obtain results with high translational relevance echoing human disorders. Given significant sex differences in stress reactivity, both male and female rodents should be assessed in order to investigate sex-specific mechanisms underlying stress reactivity. This will not only shed more light on the underlying mechanisms of stress-related disorders, but will help to develop more targeted treatment options.

## Figures and Tables

**Figure 1 brainsci-08-00121-f001:**
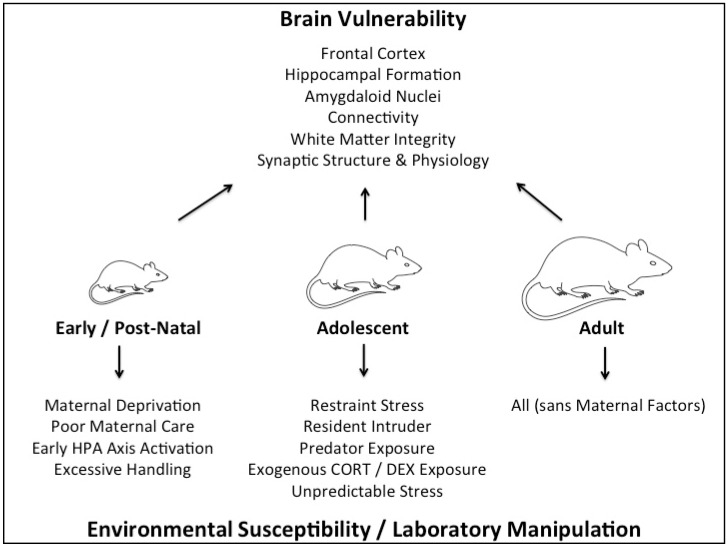
Divergence in the **Modeling of Developmental Stress.** The modeling of stress across development in rodents has been achieved using a range of different stress treatments, some which are specific to certain developmental periods (e.g., maternal deprivation during early development) while others can be applied independent of age (e.g., unpredictable stress). The length, severity, and type of stress can also be modulated to address relevant hypotheses. While “stress”, especially that of the chronic form, may intrinsically affect regions of the brain with concentrated GR expression (e.g., the hippocampus), when stress occurs within different developmental windows, divergent behavioral effects may emerge, that can have differing value for the investigation of psychiatric endophenotypes. CORT = corticosterone; DEX = dexamethasone; HPA axis—hypothalamic-pituitary-adrenal axis.

**Figure 2 brainsci-08-00121-f002:**
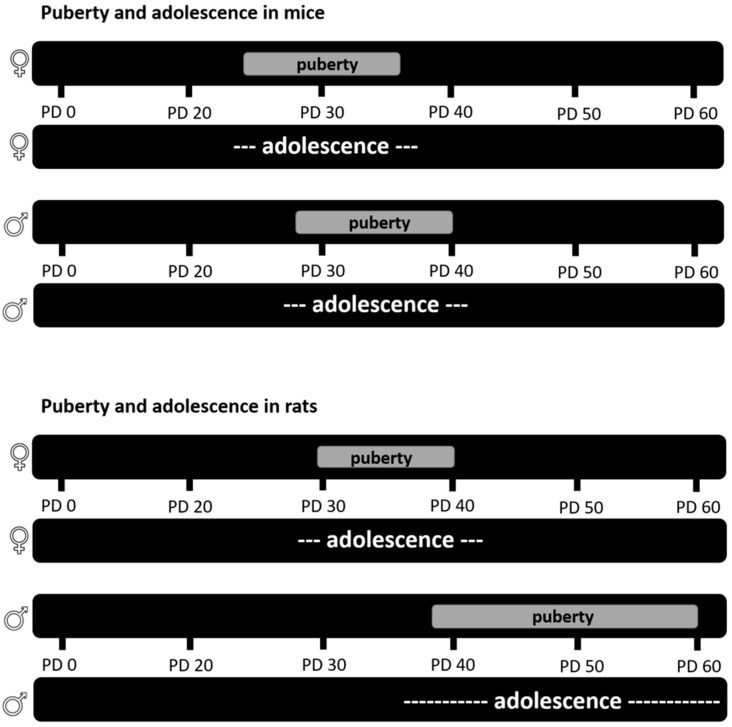
Puberty in male and female mice and rats. The timing of puberty and adolescence in male and female mice (**top**) and rats (**bottom**) (modified after [125]). PD = postnatal day.

**Table 1 brainsci-08-00121-t001:** The effects of early, adolescent, and adult stress on behavioral and molecular phenotype in adulthood.

Stress Timing	Type of Stress	Behavioral Phenotype in Adulthood	Molecular Phenotype in Adulthood
**Early life stress**	psychological	depression-like behavior, cognitive impairment [41]	HPA-axis abnormalities [41]
		memory deficits, enhanced emotional learning [42]	improved hippocampal neurogenesis [42]
		improved hippocampal-dependent memory [43]	reduced levels of hippocampal glucocorticoid and mineralocorticoid receptors [43]
		improved spatial learning and enhanced anxiety at 2 months, but spatial memory deficits and normal anxiety levels at 15 months [44]	impaired LTP under basal conditions, but increased LTP in response to high CORT [43]
		depression-like behavior in males and females with greater effect in males [45]	decreased mBDNF in the dorsal hippocampus in males only [46]
		increased preference for alcohol in males but not females [47]	increased mBDNF in the ventral hippocampus in females only [46]
**Adolescent stress**	physical and social	impaired learning behavior in radial water maze while working and spatial memory remained intact [48]	liposaccharide (LPS) induced exaggerated elevation of the pro-inflammatory cytokines IL-1β and TNF-α in males but not in females [49]
	physical	spatial memory deficits and hippocampal volume changes [50]	reduced hippocampal GR, increased hippocampal volume [51]
		spatial memory deficits [51]	
		reduced social interaction, depression-like behavior [52]	
		spatial memory deficit [53]	
	social	no spatial memory deficits or hippocampal volume changes [50]	decreased amplitude of spontaneous excitatory postsynaptic currents in the PFC only in male mice; decreased amplitude of spontaneous excitatory postsynaptic currents in the nucleus accumbens only in female mice [54]
		impaired spatial memory in Y maze and working memory in Morris water maze, while social recognition memory and episodic memory are intact [55]	
		spatial memory deficits in males and females [56,57]	
		reduced contextual fear conditioning [58]	
		anxiety-like behavior [54]	
	pharmacological	no changes in spatial memory, novel object recognition, anhedonia or anxiety in males or females (CORT 50 mg/L) [46]	deficits in sensory gating as measured by PPI only in males (CORT 50 mg/L) [59]
		no spatial memory deficit (CORT 40 mg/L)	
**Adult stress**	physical and social	depression-like/anxiety-like and submissive phenotype [60]	
	physical	anxiety-like behavior [61]	reduced immune response [62]
		depression-like phenotype and body weight loss [63]	
		deficit in recall memory (Morris Water Maze) [64]	
		anhedonia-like phenotype [65]	
		anhedonia-like and anxiety-like phenotype [66]	
		anxiety-like phenotype in defensive burying test; no anxiety-like phenotype in EPM or light/dark box [67]	
		chronic restraint stress induced anxiety-like phenotype; CUS induces anxiety-like and depression-like phenotype [68]	
		somatic effects in males and anxiety-like phenotype in females [69]	
		21 days of restraint stress enhanced spatial memory, while 28 days of restraint stress either had no effect or impaired spatial memory in females [70]	
		14 days of stress enhanced spatial memory, while 21 days of stress impaired spatial memory in males [71]	
	social	depression- and anxiety-like phenotype; impaired memory [72] anxiety-like phenotype and episodic memory deficit [73] spatial-memory impairment [74]	increased oxidative stress and inflammation [72] reduced complexity of apical dendrites of CA3 neurons [74]
	Disruptions to normal circadian rhythm (physical and psychological)	depression-like and anxiety-like phenotype [75]	reduced dendritic length in the DG and CA1 [76]
		no changes in memory or anxiety [77]	
		hippocampal memory deficit [78]	
		anhedonia-like and depression-like phenotype; learning and memory impairment [76]	
	pharmacological	depression-like phenotype [79]	reduced hippocampal neurogenesis [80,81]
		increased anxiety-like behavior [82]	increased apoptosis [33,83,84]
		depression-like phenotype in C57BL6/J but not in C57BL6/N [85]	reduced hippocampal neurogenesis in males and females [31]
		enhanced emotionality score or no effect in females [86]	no changes in cell proliferation, survival or neuronal maturation in DG of the hippocampus in females [86]

The effects of diverse types of chronic stress during early life, adolescence and adulthood in rodents are summarized. Blue color represents studies looking at only male rodents, while red color shows studies looking either at females only or at both sexes. Items in the right-most two columns are independent lists and are not linked to each other. CA1 = Cornu Ammonis 1; CA3 = Cornu Ammonis 3; CORT = corticosterone; CUS = chronic unpredictable stress; DG = dentate gyrus; EPM = elevated plus maze; GR = glucocorticoid receptor; HPA = hypothalamic-pituitary-adrenal axis; LTP = long-term potentiation; mBDNF = mature brain-derived neurotrophic factor; PFC = prefrontal cortex; PPI = prepulse inhibition.

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
