# Peer review of "On the Developmental Timing of Stress: Delineating Sex-Specific Effects of Stress across Development on Adult Behavior"

_brainsci, 2018, doi:10.3390/brainsci8070121_

Round 1

Reviewer 1 Report

The article reviews different types of stress-induced neurological changes at three time points - early life, adolescence and adulthood in (mostly) male rodents. The article serves as a literature review. The authors try to argue that the brain changes noted as a consequence of such stress might result in neuropsychiatric disorders like schizophrenia and depression.

The paper is well organized as far as age of occurrence and type of stressor are concerned, but I see no real compelling argument that the multitude of changes may or may not lead to a brain disorder like schizophrenia in humans. There is a neurodevelopmental hypothesis of schizophrenia which, updated, might include some of these effects, but it's difficult to know - the effects are SO different depending on stressor type, rodent type and age it's almost impossible to come up with any "centralized" argument(s). And there is no evidence cited that the effects seen in rodents mirror the brain damage seen, e.g., in schizophrenia in humans.

There is a lack of physiological stress research in female rodents AND the stress response is different, as noted. It is also noted that depression is more common in females. The authors report on a few studies that result in a few neural differences depending on the timing and type of stressor, but again offer no compelling evidence that such a response in male rodents has any predictive importance for depression in humans.

My opinion is that the paper should be rewritten as simply a review paper - the effects of different stress types on different aged rodents - perhaps a "stress" paper specifically devoted to neural changes in rodents. At this point there is simply not enough evidence to support the idea that changes reported in rodents, IF also occurring in humans, result in neuropsychiatric disorders. I would vote to reconsider following a major revision of the paper in this format.

Author Response

Response to Reviewer 1 Comments

 1.      The paper is well organised as far as age of occurrence and type of stressor are concerned, but I see no real compelling argument that the multitude of changes may or may not lead to a brain disorder like schizophrenia in humans. […] the effects are so different depending on stressor type, rodent type and age it’s almost impossible to come up with any “centralized” argument(s). And there is no evidence cited that the effects seen in rodents mirror the brain damage seen, e.g., in schizophrenia in humans.

A section has been added to emphasise the heterogeneousness of stress-induced behavioural and neurological changes and that it would be previous to equate findings from rodents to aetiology in human disease, but that rodent models offer a controlled tool to delineate specific mechanisms of stress. For example, the following is added to line 131:

Given the complex nature of different types of stressors and the body’s differing responses to them, which are themselves compounded by a myriad of factors including genetics, environment (prenatal and postnatal) and sex, the study of stress and its fundamental effects in humans is extremely difficult. Rodent model is a tool that allows us to isolate and examine specific types of stressors at specific time points of development while minimising genetic and environmental differences. This will allow us to uncover some important mechanisms of how stress can influence brain functions, which may offer knowledge important towards the treatment of human neurological disorders that are linked to stress.

Elsewhere, direct comparisons to human disorders have been removed. The language used in places have also been softened to avoid direct comparison to human disorder, for example, in line 270, when discussing the forced-swim test, the test has been altered to be described as inducing ‘behavioural despair’ instead of ‘depression-like behaviour’.

 2.      There is a lack of physiological stress research in female rodents and the stress response is different, as noted. It is also noted that depression is more common in females. The authors report on a few studies that result in a few neural differences depending on the timing and type of stressor, but again offer no compelling evidence that such a response in male rodents has any predictive importance for depression in humans.

Given the heterogeneity and paucity of current data regarding differences in stress response between male and females, one of the points we wish to stress in this manuscript is the need for future research to be conscious of including female animals. We have tried to emphasise this further. For example, the following is added to line 145:

[…] and the importance for future studies to differentiate and explore differences between the sexes, whereby important mechanistic insight may be gained.

And to line 173:

Furthermore, the evidence underline the fundamental role sex plays in mediating stress-induced effects and advocates for the importance to include both sexes in rodent studies.

 3.      My opinion is that the paper should be rewritten as simply a review paper – the effects of different stress types on different aged rodents – perhaps a “stress” paper specifically devoted to neural changes in rodents. At this point there is simply not enough evidence to support the idea that changes reported in rodents, if also occurring in humans, result in neuropsychiatric disorders.

We have made considerable efforts to edit the manuscript, as the reviewer suggested, to avoid drawing direct implications from the rodent model findings to human disorders. Much references to human disease has been deleted (for example, from line 421, where we had inferred validity of the CUS model to psychiatric diseases and from line 513 onwards in the Adult Stress section, among others) to refocus the review purely on behavioural and neurological changes in rodent models in response to various types of stressors at various developmental stages.

Reviewer 2 Report

Re: brainsci-317022

Schroeder et al have taken on the ambitious task of reviewing developmental timing of the effects of stress.  It is well written, and comprehensive.  The main issue is that it could benefit from a bit more statement of awareness around the periphery of the covered areas.

For one example, the HPA axis is stated as if it were the only stress response system.   The sympathetic adrenal medullary system is also part of this response.  The only mention of this that I could find was a passing statement on line 382.  Admittedly, it is not clear how this might impact longer lasting stress effects, but it would be good to present an awareness of this.  Also, it becomes of some relevance when discussing the effects of stress on cognition (example, the effect of social stress on problem solving tasks, reversed by the noradrenergic antagonist propranolol (Alexander et al J Cogn Neurosci 2007;19:468-478)).  This cognitive aspect seems to best fit in the first part of the Adult section.

Second, a little more presentation of awareness of prenatal stress effects seems worthy of brief mention, expanding on the passing mention at the beginning of Section 2.  A recent review covers this, along with genetic components (Abbott et al Psychoneuroendocrinology 2018;90:9-21), to allow brief comment on how this is a separate topic with its own set of implications (and also for potential effects of genes on periconception environmental stressors mentioned on lines 208-210).

For the effects of adolescence, it would be good to show awareness of this time period being potentially important for transgenerational effects, as demonstrated by a number of studies in Tracy Bale’s lab, in addition to examples of potential clinical salience (such as Roberts et al JAMA Psychiatry 2013;70:508-515).

Finally, it seems worthy of brief mention an awareness of other genes that are important.  The most obvious one is the serotonin transporter protein, where there is a substantial literature on its role in the relationship between stress exposure and psychiatric outcomes.

There are other more minor issues:

Table 1- might clarify in the legend that these columns are lists and not linked.  At the beginning of the table, I was struggling to line up the two right-most columns before I realized that wasn’t the intent.  Also, in the Table section under Adult stress- physical, it seems many of these items could be grouped for efficiency in the Behavioral column.  It looks redundant as it is presented.

The reference call outs from the text were odd.  For example in line 163 ‘Oomen, Soeters’- there were many other examples.  Perhaps this is a requirement for this journal, but if not, it appears odd.

On line 260 would be helpful to provide a brief example of the bases for ‘many claim to distinguish an adolescent period in rodents based on changes…’.

Lines 638-639- please clarify or rephrase, it seems that ‘upregulation of 219% compared to controls’ does not fit with ‘subtle disruptions in GR gene expression’.

There were a number of typographical errors, most noticeable in the Adult section.  Line 439 change ‘worker’ to ‘workers’, line 483 change ‘weeks’ to ‘week’, line 487 change ‘restrain’ to ‘restraint’, line 506 change ‘appear’ to ‘appears’

Author Response

Response to Reviewer 2 Comments

 1.      The sympathetic adrenal medullary system should be mentioned as part of the stress response.

The following has been added to line 46, in the Introduction section:

 The acute stress response, or ‘fight and flight’ response, was first described by Walter Bradford Cannon early in the 20th century. Perceived stressors which challenged “homeostasis”, a term Cannon coined, activate the sympatho-adrenomedullary system, resulting in the immediate release of catecholamine. As the name suggests, the immediate effect of the acute response is to marshal the body’s resources to combat, or flee from, the immediate source of stress by elevating heart rate, releasing stored energy to muscles, dilating blood vessels to muscles and boosting metabolic rate [5]. In the brain, noradrenaline increases arousal and attention as well as mediates memory formation and retrieval [6]. While noradrenaline has been shown to be necessary for attentional set-shifting [7], elevated level has been shown to impair lexical-semantic and associative network flexibility [8]. Adaptation of the sympatho-adrenomedullary system occurs if the same stressor is repeatedly and predictably experienced but upon novel stressors, the system revert to high release of catecholamine [9, 10]. Importantly, evidence show that this acute stress response is influenced by the slower central stress response system, the Hypothalamic-Pituitary-Adrenal (HPA) axis [11]. Given the prevalent role of the HPA axis in regulating stress adaptation, its dysfunction has been heavily implicated in a wide variety of disorders [12].

 2.      Also, it becomes of some relevance when discussing the effects of stress on cognition (example, the effect of social stress on problem solving tasks, reversed by the noradrenergic antagonist propranolol (Alexander et al J Cog Neurosci 2007)).

We have added a section in the paragraph mentioned above for point 1, briefly exploring the noradrenergic influences on cognition, including the reference the reviewer has suggested among others.

 3.      More presentation of awareness of prenatal stress effects seems worthy of brief mention.

The following has been added to line 165 of the Early Life Stress Model section:

While it being beyond the scope of the current review, it is important to note that stress begin affecting the organism in utero. Despite being in the relatively cocooned environment of the womb, there is complex interplay between maternal genetics, offspring genetics, timing of stress and the nature of stress, which together dictate the influence on offspring outcomes [46, 47]. Prenatal stress can greatly impact upon the fetus and lead to permanent modification of the HPA axis and stress responses later in life in a sex-dependent manner [46, 48]. These evidence point to the significant role stress has, especially during critical phases of brain development, on the long-term function of the organism. Furthermore, the evidence underline the fundamental role sex plays in mediating stress-induced effects and advocates for the importance to include both sexes in rodent studies.

 4.      For the effects of adolescence, it would be good to show awareness of this time period being potentially important for transgenerational effects, as demonstrated by a number of studies in Tracy Bale’s lab, in addition to examples of potential clinical salience (such as Roberts et al JAMA Psychiatry 2013; 70:508-515).

The following has been added to line 305 of the Adolescent Stress Model section, including the reference the reviewer has suggested among others:

Furthermore, it has been shown that periadolescent stress can affect stress responsiveness of the individual during and after pregnancy [83], thereby having the potential to affect stress adaptation in a transgenerational manner, through both behavioural and epigenetic modulation [84, 85]. For example, maternal exposure to early life abuse has been linked to an increased risk in the offspring to various mental disorders such as autism [86] and ADHD [87].

 5.      It seems worthy of brief mention an awareness other genes that are important. The most obvious one is the serotonin transporter protein, where there is a substantial literature on its role in the relationship between stress exposure and psychiatric outcomes.

The following has been added to line 686 of the Genetically Modified Animal Models and the Study of Stress section:

Examples include a polymorphism (5-HTTLPR) within the promoter region of the serotonin transporter gene, where a short allele, with a 43bp deletion, results in hypersensitivity of the HPA axis [184-186]. Another example is the single nucleotide polymorphism (rs53576) in the oxytocin receptor gene, which has been shown to modulate HPA axis stress responses [187].

 6.      Table 1 – clarification of organisation of data and grouping of Adult stress-physical references.

The following has been added to the Table legend:

Items in the right-most two columns are independent lists and are not linked to each other.

Furthermore, in the Adult stress-physical section, the body weight loss phenotype (ref 141) has been shifted to the behavioural column.

 7.      The reference call outs from the text were odd.

These have been edited to more standard formats.

 8.      One line 260 would be helpful to provide a brief example of the bases for ‘many claim to distinguish an adolescent period in rodents based on changes…’.

The following has been added to line 322 in the Adolescent Stress Model section:

[…] in behaviour, such as increased risk taking; neuronal development in the frontal-cortical and limbic brain regions; and changes in gonadal hormone levels [45],

 9.      Line 638-639- please clarify or rephrase, it seems that ‘upregulation of 219% compared to controls’ does not fit with ‘subtle disruptions in GR gene expression’.

The section on line 737 has been rephrased to the following to more accurately reflect the data:

In this study, GR heterozygote mice showed a downregulation of GR mRNA to 33% of littermate controls while GRYGR mice showed upregulation of GR mRNA to 219% of littermate controls [197], which suggests that, based on the relative difference in gene expression between the two models, disruptions to GR gene expression has the ability to shape behavioural processes.

 10.  There were a number of typographical errors, most noticeably in the Adult section.

We have made a concerted effort to proof-read the manuscript and corrected a number of typographical and grammatical errors.

Round 2

Reviewer 1 Report

The edits made were helpful in reducing the implicit suggestion that neural alterations resulting from stress in rodents paralleled results leading to human psychological disorders (of which there is no data). I added further suggestions below  - the paper is in significant need of editing before it may be published - e.g., it's "catecholamines, not catecholamine" every time. My comments focus on the intro primarily  -I didn't make any comments/changes to the results of the studies mentioned, and the conclusion was cleaned up appropriately -

In the abstract, line 15 - "...this may contribute ..."

line 29, omit the word "here"

in the introduction, omit lines 36-42 - all mention of Selye - after "...neuroendocrine systems" the next line should start at 43. If you're going to mention him you need to mention why he was incorrect on #3, and it takes too much time.

add the word "also" to the end of the line "...the stress response is also" ...

line 63 - "the HPA axis modulates a multitude..."

line 83 - "...important findings that may help contribute to explaining why ..."

line 87 - Interestingly, despite the well-described ...male rodents only to avoid female estrous cycle confounding. END THE PARAGRAPH

line 97 end the sentence at "cortex" Start the next sentence with "Excitatory feedback... on line 99.

line 114 - start a new paragraph here

line 171 - "results" not "evidence"

line 510 - end the sentence as written with "adulthood. Omit all until line 530 "Despite ..." Omit all until line 548.

Author Response

The edits made were helpful in reducing the implicit suggestion that neural alterations resulting from stress in rodents paralleled results leading to human psychological disorders (of which there is no data). I added further suggestions below  - the paper is in significant need of editing before it may be published - e.g., it's "catecholamines, not catecholamine" every time. My comments focus on the intro primarily  -I didn't make any comments/changes to the results of the studies mentioned, and the conclusion was cleaned up appropriately -

References to ‘catecholamine’ has been changed to ‘catecholamines’.

In the abstract, line 15 - "...this may contribute ..."          

This has been corrected.

line 29, omit the word "here"

This has been corrected.

in the introduction, omit lines 36-42 - all mention of Selye - after "...neuroendocrine systems" the next line should start at 43. If you're going to mention him you need to mention why he was incorrect on #3, and it takes too much time.

Reference to Selye on lines 36-42 has been removed.

add the word "also" to the end of the line "...the stress response is also" ...

This has been added (line 58)

line 63 - "the HPA axis modulates a multitude..."

This has been edited as suggested.

line 83 - "...important findings that may help contribute to explaining why ..."

This has been edited as suggested.

line 87 - Interestingly, despite the well-described ...male rodents only to avoid female estrous cycle confounding. END THE PARAGRAPH

The following has been added to line 88, with the rest of the paragraph removed as suggested:

to avoid female estrous cycle being a confounding factor.

line 97 end the sentence at "cortex" Start the next sentence with "Excitatory feedback... on line 99.

This has been edited as suggested (beginning line 98).

line 114 - start a new paragraph here

This has been done (line 115).

line 171 - "results" not "evidence"

This has been corrected (line 173).

line 510 - end the sentence as written with "adulthood. Omit all until line 530 "Despite ..." Omit all until line 548.

This edit has been done as suggested (starting line 512).